# Nanoscale Calculation of Proton-Induced DNA Damage Using a Chromatin Geometry Model with Geant4-DNA

**DOI:** 10.3390/ijms23116343

**Published:** 2022-06-06

**Authors:** Kun Zhu, Chun Wu, Xiaoyu Peng, Xuantao Ji, Siyuan Luo, Yuchen Liu, Xiaodong Wang

**Affiliations:** 1School of Nuclear Science and Technology, University of South China, Hengyang 421001, China; zhuk@stu.usc.edu.cn (K.Z.); pengxy@stu.usc.edu.cn (X.P.); 20202010210608@stu.usc.edu.cn (X.J.); 20192010110390@stu.usc.edu.cn (S.L.); liuyuchen@stu.usc.edu.cn (Y.L.); 2School of Nursing, University of South China, Hengyang 421001, China; 2014002037@usc.edu.cn

**Keywords:** Monte Carlo simulation, Geant4-DNA, DNA damage, uncertainty analysis

## Abstract

Monte Carlo simulations can quantify various types of DNA damage to evaluate the biological effects of ionizing radiation at the nanometer scale. This work presents a study simulating the DNA target response after proton irradiation. A chromatin fiber model and new physics constructors with the ELastic Scattering of Electrons and Positrons by neutral Atoms (ELSEPA) model were used to describe the DNA geometry and the physical stage of water radiolysis with the Geant4-DNA toolkit, respectively. Three key parameters (the energy threshold model for strand breaks, the physics model and the maximum distance to distinguish DSB clusters) of scoring DNA damage were studied to investigate the impact on the uncertainties of DNA damage. On the basis of comparison of our results with experimental data and published findings, we were able to accurately predict the yield of various types of DNA damage. Our results indicated that the difference in physics constructor can cause up to 56.4% in the DNA double-strand break (DSB) yields. The DSB yields were quite sensitive to the energy threshold for strand breaks (SB) and the maximum distance to classify the DSB clusters, which were even more than 100 times and four times than the default configurations, respectively.

## 1. Introduction

Proton therapy has the benefits of providing a therapeutic dose to tumors with significantly lower dose delivery to normal tissues [1]. This therapy is receiving increasing attention in clinical cancer treatment. DNA is considered the most sensitive target for most the biological effects of ionizing radiation [2].

Many studies have focused on DNA single-strand breaks (SSBs) and double-strand breaks (DSBs), the latter of which are the most common lesions caused by irradiation [3,4,5]. Experiments in radiobiology, such as pulsed-field gel electrophoresis and H2AX analysis [6,7,8,9], examine DNA behavior to study strand breakage. However, these methods do not allow for direct assessment of the number of strand breaks or distinguishing between DSBs and DSB clusters [10,11]. Thus, we used a mechanistic approach with Monte Carlo Track Structure (MCTS) simulation is to quantify the various types of DNA damage by focusing on the nanoscale modeling of DNA molecules and the interaction of water radiolysis within a time scale of femtoseconds (10^−15^ s) to microseconds (10^−6^ s). 

Several MCTS codes have been developed for radiobiological research, such as KURBUC [3,11,12,13,14,15,16], PARTRAC [17,18,19,20,21,22], NASIC [23,24,25] and RITRACKS [26,27,28,29], but they are not open sources. Geant4-DNA [30,31,32], a low-energy extension of Geant4 [33,34], is free to access and include accurate physics processes at a scale relevant for DNA models.

Indeed, the distance between neighboring ionizations in the track of a heavily ionizing particle is in the order of nanometers, which is the same order of magnitude as the size of the many molecules that make up DNA. Hence, a realistic geometrical model of DNA targets is an important aspect in MCTS simulations for DNA damage evaluation. The DNA, nucleosome, and chromatin fiber were originally depicted as cylinders with a specific diameter [35]. Subsequently, Charlton, Pomlun and Pinak et al., created more realistic DNA models accounting for the base and sugar-phosphate backbone as well as their position and rotation [36,37,38]. Since Michalik proposed the detailed geometry of nucleosomes, Pomlun and Nikjoo et al., built their own models of nucleosomes [39,40,41]. Friedland et al., have constructed a model in PARTRAC with five levels of structure from DNA to the chromatin fiber circle on the basis of Holley’s chromatin fiber model. Then, using various technologies such as fractal geometry, dnaFabric software and pdb files, more precise models of human fibroblast nuclei were created [42,43,44,45,46,47,48,49,50]. The DNA atomic model is more realistic and accurate than prior models, ranging from simple cylinders to highly sophisticated multiscale models. Although several atomic DNA models have been described and analyzed in the literature described above, they are not entirely open-source or free to use.

To accurately evaluate the DNA damage with Monte Carlo methods, investigating the uncertainty of DNA damage simulation is an important scientific problem. A relevant aspect of MCTS simulations is the precision of physical processes at a scale suitable for DNA models. Indeed, creating the currently available cross-sections of the liquid water medium presumed in clinical radiotherapy for charged particle track simulation was an enormous task, and the cross-sections were only recently made widely available to the broader scientific community via the Geant4-DNA toolkit [51]. In Geant4-DNA, three recommended physics constructors were used, each including all the required lists of particles, physics processes and associated models for Geant4-DNA simulation application [52]. These constructors, “G4EmDNAPhysics_option2” (option2) [53], “G4EmDNAPhysics_option4” (option4) [54] and “G4EmDNAPhysics_option6” (option6) [12], have the same set of models for protons in liquid water. Secondary electrons are responsible for the majority of direct DNA damage caused by electron interactions: Option2 does not include relativistic corrections and shows disagreement with the experimental data at low energy and intermediate angles [55]. The energy thresholds of the elastic scattering process for Option4 and Option6 are merely 10 keV and 256 keV, respectively. To resolve this issue, Shin et al., have developed the “ELastic Scattering of Electrons and Positrons by neutral Atoms” (ELSEPA) model into Geant4-DNA to extend the elastic scattering cross-section to 1 GeV and include relativistic corrections. Geant4-DNA combined the G4DNACPA100ElasticModel with the G4DNAChampionElasticModel into a new physics constructor G4EmDNAPhysics_option8 (option8) [56]. Similarly, Lund et al., have developed a physics constructor called G4EmDNAPhysics_hybrid2and4(hereafter “option2and4”) to extract the best features of option2 and option4 [57,58] in TOPAS-nBio [59,60], which is an extension to the TOPAS [61] Monte Carlo application based on Geant4. All their work was aimed at developing a recently advanced elastic model that would perform better than any of Geant4’s electromagnetic (EM) models. Simplified descriptions of physical reality and empirical parameters are introduced in the simplified model as part of the MC simulation process. The simulation results may be influenced by parameter values.

In this work, the Geant4-DNA simulation toolkit was used to investigate the DNA response after proton irradiation. A solenoid chromatin fiber model was implemented in Geant4-DNA for early DNA damage simulation. The ELSEPA model substituted the default elastic scattering model of three recommended physics constructors available in Geant4-DNA, denoted option2 with ELSEPA, option4 with ELSEPA and option6 with ELSEPA. The “option2and4” was also included in Geant4-DNA.We compared them with three recommended Geant4-DNA physics constructors, using dedicated Geant4 examples. The yields of SSBs and DSBs were calculated with these physics constructors through comparison with data from published simulations and experimental measurements. Furthermore, the impact of the physics constructors, the energy threshold model for SSBs and the clustering distance for DSBs on the yield of DSBs and the ratio of SSBs to DSBs induced by monoenergetic protons was studied to estimate the uncertainties of DNA damage calculation.

## 2. Results

### 2.1. Benchmarking the Physics Constructors

Figure 1 shows results for the MFP simulations with various physics constructors in liquid water. For the total MFP value, opt2and4 was nearly equal to that of option2 and option4 from 10 eV to 10 keV (Figure 1a,b). Option2 and option4 including the ELSEPA model have a lower total MFP value than that for option2 and option4 in Geant4-DNA in the entire energy region. For the inelastic MFP value, we noticed that opt2and4 is the closest to the result simulated by Emfietzouglou and the option4 with ELSEPA is the closest to the measured data by Ashley below 100 eV. Except for option6 with ELSEPA, other physics constructors simulated for the inelastic MFP value in this work were consistent with the referenced data [62,63].

Figure 2 shows the track length and the penetration calculated with different physics lists. Figure 2a shows the total path length of electrons in liquid water from 10 eV to 1 MeV, which was calculated with different physics constructors listed in Table 1, with the continuous-slowing-down-approximation (CSDA) method and other MC simulations. Option2 and option2 with ELSEPA had the maximum track length, whereas the total path length of option6 was the lowest among the physics lists from 10 eV to 2 keV. The track length for option2 sharply increased above 300 eV and was linear on a logarithmic scale, and it rose slowly between 45 and 300 eV. It is clear from Figure 2b, two minima were observed below 45 eV, at 25 eV and 35 eV, respectively. The results for the other physics constructors showed the same pattern as that for option2. The higher the energy, the smaller the difference between the different physics constructors. Figure 2a also shows that the results for options with ELSEPA had the same values as those for default options in Geant4-DNA except for option6, whose track length was slightly less than that of option6 with ELSEPA. Furthermore, the track length of option2and4 is close to that of option4. Of note, our results agreed well with other MC simulations and the CSDA track length above 2 keV.

Figure 3 compares the dose-mean linear energy y_D_ by various physics constructors from 10 eV to 10 keV at different scales. As shown in the Figure 3a, the value for option6 was highest in the entire energy range, rapidly rising from 10 eV, before peaking and then decreasing. Option6 with ELSEPA also had a larger y_D_ than those of the other options. Clearly, the values obtained for other options were as close to each other as their inelastic cross-sections; notably, those for option2and4 and option4 were nearly identical. For the different scales of scoring volume (Figure 3a−c), the different physics constructors used had the same tendency as a function of initial electron energy, and the larger the scoring volume, the closer the agreement. Independently of the physics list used, the maximum yD was found at ≈300 eV for the 2.3 nm sphere, at ≈500 eV for the 10 nm sphere and at ≈1 keV for the 30 nm sphere.

### 2.2. DNA Damage Simulations

We compared the total SBs (SSBs + 2DSBs), the DSB yields, the SSB yields and the ratio of DSB to SSB induced by monoenergetic protons with other published results in Figure 4. Even though each study employed a different DNA model and varied irradiation circumstances, all the findings in this work are of the same order of magnitude and do not change much with energy. Similar to the results simulated by Sakata et al. [70] and PARTRAC [19], the TSBs yield (Figure 4b) and SSB yield (Figure 4c) increase with increasing initial proton energy, although they are smaller than those of PARTRAC and Sakata et al., However, the order of magnitude of SSB yield is not reaching the level of the experiment SSB yield of plasmid by Leloup et al. [7]. As shown in Figure 4d, our results for DSB yield were consistent with those simulated by Sakata et al., (above 2 MeV) and were close to the experiment data obtained by electrophoresis by Frankenberg et al. [71], especially at 5 MeV. The SSB/DSB ratio (Figure 4a) increases as a function of initial proton energy.

### 2.3. Parameter Sensitivity Simulations of DNA Damage

Dependencies of the ratio of DSB yields to SSB yields and DSB yields on the physics constructors are shown in Figure 5. In general, higher initial proton energies increase the ratio of SSBs to DSBs (Figure 5a) and the yield of SSBs (Figure 5b), regardless of physics constructors used, whereas the yield of DSBs decreases (Figure 5c). Of note, the opt2and4 and option4 give the equivalent ratio of SSB to DSB, the yield of SSBs and DSBs. For all the physics constructors used this work, option6 gave the highest DSB yields and lowest SSB/DSB ratio while option2 including the ELSEPA model had the lowest DSB yields. Considering the influence of elastic scattering model on the simulation, in contrast with the SSB yield results showing less difference, the DSB yields of all default options are higher than the ELSEPA model.

The influence of the ET model for the breaks in Figure 6 clearly shows that the results of strand break yields were highly sensitive to the ET model. For the constant model, when increasing the energy threshold value, both the DSB yields (Figure 6a) and ratio of DSB yields to SSB yields (Figure 6b) were reduced. However, DSBs decreased sharply from 8.22 to 17.5 eV (Figure 6d), whereas DSBs had little changes in the whole energy range between 17.5 and 21.25 eV. Figure 6 also shows that the results for the linear proportional model appeared to be consistent between the 12.6 and 17.5 eV constant model.

Figure 7 shows the effects of varying the clustering distance on the DNA damage results. Changing the clustering distance resulted in a different total yield of DSBs (Figure 7b) and different ratio of the yield of DSBs to the yield of SSBs (Figure 7a). The clustering distance of 40 bp gave the highest total yield of DSBs, whereas 3 bp gave the lowest DSBs yield. Figure 7d shows that 40 bp and 30 bp caused up to 147% and 111% more DSBs, respectively, than 10 bp, and 3 bp caused up to 139% less DSBs than 10 bp. Figure 7b shows that a different clustering distance caused the similar ratio of DSBs yield to SSBs yield at high proton energy.

## 3. Discussion

This study aims to calculate the proton-induced DNA damage and understand how the parameters of scoring the DNA damage model in MCTS simulation in a physical stage affect the DNA SSB and DSB yields. In the physical stage of water radiolysis, most of the DNA damage is caused by secondary electrons. We first verified the establishment of the extended physics constructors and studied its influence on the fundamental MCTS simulation quantities and the microdosimetry spectra y_D_. It was found that from Figure 1a,b, the inelastic models affect the total MFP value because opt2and4 has the same total MFP value as option2 and option4 from 10 eV to 10 keV, which have the same inelastic models listed in Table 1. For the inelastic MFP value, opt2and4 is closest to the result simulated by Emfietzouglou et al. [63]. Because the contribution of elastic scattering was small at high energy, the differences between the physics lists used became smaller with increasing initial electron energy. For the dose-mean linear energy simulation, option6 with ELSEPA also had a larger y_D_ than the other options, owing to the larger inelastic cross-sections of the CPA100 model. Overall, due to the major contribution of elastic scattering, MFP, track length, and y_D_ are all affected by elastic scattering at energies below 100 eV.

DNA damages caused in the physical stage were quantified as the total DNA damage yield in this work. Because DSB is made up of two SSBs, the trend of SSB and DSB as a function of proton energy is indeed opposite. It is still a challenge when compared to experiment data because of the uncertainties of measurement due to cell type, measurement condition, beam quality, and other factors. Compared with the data of Leloup et al., (Figure 4c,d), the DSB yields and SSB yields are higher than other data, which is apparently due to the influence of base pair density and histone scavenging effect [70]. The base pair densities of the plasmid used by Leloup et al., the hamster cell (V79) used by Frankenberg and the human fibroblast cell in this work are 9.4 × 10^−6^, 0.015, and 0.012 bp/nm^3^, respectively. Thus, the DSB yields simulated in this work match well with the data measured by Frankenberg et al., (Figure 4d) within a 9% difference. The further studies are required for simulations of scavengeable damage for protons [70].

Changing physics constructors resulted in a different ratio of SSB to DSB and a different DNA damage yield of DSBs and SSBs. Figure 5b shows that option4, option6, opt2ELSEPA, opt4ELSEPA, opt6ELSEPA and opt2and4 cause 2.59%, 6.92%, 3.31%, 3.25%, 6.976% and 3.1% difference in the SSB yields, respectively, compared with option2, whereas Figure 5c shows that the yield of DSBs was changed 19.45%, 56.4%, 14.6%, 13.35%, 47.82% and 20.38% compared with option2. Hence, the ratio of SSB to DSB had the opposite trends to the DSB yields (Figure 5a). The DSB yields were more sensitive to the physics constructors than the SSBs yield; this is consistent with the fact that elastic scattering does not affect the energy deposition but affects the distribution of particles in space. Compared with the default physics constructor including the ELSEPA model, the ELSEPA model can cause 14.594%, 10.2% and 9.16% difference for option2, option4 and option6, respectively. The DSB yields of all default options listed in Table 2 are higher than the ELSEPA model, because the default options are less diffusive than the options with the ELSEPA model. Thus, option6 and option6 including the ELSEPA model gave the highest DSB yields related to the highest interaction cross-sections in the CPA100 model than the other models.

Calculations of SSB and DSB yields were highly sensitive to the DNA damage ET model. For the constant energy threshold model, DSB yields can vary more than 100 times when the energy threshold changed from 8.22 to 17.5 eV in our study. Of note, the DSB yields had little changes in the entire energy range with the ET value from 17.5 to 21.25 eV. This is because most energy deposit events have energy below 17.5 eV. Compared with the linear proportional model used in PARTRAC, the DSB yields appeared to be consistent with an ET value between 12.6 and 17.5 eV constant model, which is consistent with Lampe’s solution [73] based on the calculation of the direct damage of DNA changes for electrons. 

Changing to the clustering distance resulted in different total yield of DSBs and different ratio of the yield of DSBs to the SSBs yield. In our study, the clustering distance of 40 bp and 30 bp can cause up more than two times DSB yields than the clustering distance of 10 bp. The main reason for the differences when increasing the clustering distance is the clustering algorithm. As the clustering distance increases, lesions that normally are treated as isolated lesions would be combined to form new clusters. Although adjacent clusters that would normally be classified as separate clusters would be merged, the former effect must be dominant. In the whole energy range, the DSB yields will increase as the clustering distance increases, especially for the low-energy proton.

## 4. Materials and Methods

### 4.1. Models of DNA Geometry

A chromatin fiber model was built with Geant4-DNA version 10.4, including three levels of structure from the DNA double helix to the chromatin fiber. On the basis of the geometry in Henthron et al. [74], both the sugar-phosphate backbone and base are separate volumes in our model (Figure 8a). The backbone was a quarter-cylinder with a complete radius of 1.15 nm and a deleted section for the base, whereas the base was a half-cylinder with a diameter of 1 nm. The nucleotide was 0.34 nm thick and rotated 36 degrees for each consecutive pair.

The nucleosome (Figure 8b), which was composed of 1.65 left-handed turns of the B-DNA double helix wound around a histone represented as a cylinder with a radius of 3.3 nm and a height of 5.7 nm, is the basic unit of chromatin fiber.

On the basis of Finch and Klug’s hypothesis [75], the solenoid chromatin fiber was constructed with repeating nucleosomes (Figure 8c). To ensure a seamless connection between linker ends and histone DNA, we used a path followed by the Bezier curve instead of creating the linker DNA (H1). The fiber’s diameter and length were fixed to 37 nm and 161 nm, respectively. The nucleosome helix was created around the fiber’s central axis and was programmed to repeat every 6 nucleosomes per turn (Figure 8b). The fiber was made up of 61 nucleosomes with a total of 10.8 kbp of DNA.

Chromatin fiber had a density of 4.2 nucleosomes per 11 nm, which corresponded to a loose fiber. Because liquid water makes up more than 70% of the biological medium in the human body, the total volume material was G4_WATER with a density of 1 g/cm^3^, except for the nucleotide volumes, which had a density of 1.407 g/cm^3^ in this simulation, according to the scaled cross-sections [4]. Taking into account the shape and mean size characteristics of the irradiated nuclei, a cylinder with an ellipsoidal base (semi-axes a = 9.85 μm and b = 7.1 μm, and a height h = 2.5 μm) was constructed to mimic the human fibroblasts (about 6 × 10^9^ bp) and impinged with random directions.

### 4.2. Physics Settings

The energy deposition of primary and secondary particles, primarily the latter, causes direct DNA damage. Geant4-DNA provides three sets of alternative physics model for the simulation of electron interactions with liquid water, which have been described in detail in the previous publications [52,53,54]. As indicated in Table 2, each of the three proposed physics lists (option2, option4 and option6) of Geant4-DNA for water radiolysis had the unique electron transport process, including ionization, excitation, elastic scattering, molecular attachment and vibrational excitation.

Option4 used models based on the Emfietzouglou parameterization of the dielectric response function of liquid water to provide larger and more realistic excitation cross-section data than data in the gas phase as well as more accurate low-energy corrections, particularly for exchange and correlation in electron–electron interactions, than option2. Option6 is a complete port of the well-known CPA100 track structure code, which employs the relativistic Binary-Encounter-Bethe technique for ionization modeling and the Dingfelder model of liquid water dielectric response for excitation. Option4 employs an improved screened Rutherford model with a screening factor derived from vapor water data, whereas option6 employs the traditional Independent Atom Method. In general, option4 has a more accurate W-value (the mean energy required to generate an ion pair) and dose point kernels.

The ELSEPA model developed by Shin et al., to improve the accuracy of electron elastic scattering in Geant4-DNA, which used the Dirac partial wave approach including relativistic corrections and the optical parameters included in the correlation–polarization and in the absorption potentials [55]. In this work, option2 with ELSEPA, option4 with ELSEPA and option6 with ELSEPA were used to simulate the transport and interactions, and only the elastic scattering process was replaced by the ELSEPA model. Option2and4 was associated with both option2 and option4 like option 8 combining the Champion model [53] with the CPA100 model [76].

As shown in Table 1, the energy threshold for all elastic scattering, ionization and ex-citation processes of these physical constructors was increased to 1 MeV or greater. More-over, the molecular attachment and the vibrational excitation of option2 were included in this physics constructor as in option8. These models applied the default tracking cut-off in which the kinetic energy of electrons is locally deposited.

### 4.3. The Examples of Geant4-DNA

#### 4.3.1. The “mfp” Example of Geant4-DNA

As explained by Emfietzoglou et al. [63], the “mfp” example in Geant4 simulates the mean free path values (MFP), which represents the mean distance between two interactions and is a crucial component of MCTS simulation. The total MFP value was obtained in this study to compare the option with ELSEPA with the option in Geant4-DNA, and the inelastic MFP value was calculated to compare with reference data [62,63] in the energy range of 12 eV (higher than 11 eV for option6) to 50 keV.

#### 4.3.2. The “Range” Example of Geant4-DNA

In the “range” example, three spectra were calculated: “track length”, which represented the path length of particle trajectories; “penetration,” which was the distance between the initial and final states; and “projected range”, which represented the projection along the incident direction. To verify the simulations, the track length and the penetration were estimated with various options listed in Table 1 and compared with referenced data [64,65,66,67,68,69].

#### 4.3.3. The “Microyz” Example of Geant4-DNA

The “microyz” example [52] simulated two important microdosimetry characteristics, which have been fully described by Keller [77]. One used type of microdosimetric spectra was the frequency-mean lineal energy *y_F_* defined as:(1)$$y_{F}=\int_{0}^{\infty }yf\left(y\right)\;dy,$$
where $f\left(y\right)$ is the possible density function of the linear energy *y*. The linear energy *y* is defined as the energy transfer in a single event in the target of the mean chord length: (2)$$y=\frac{\varepsilon }{\bar{l}}$$
where ε is the energy transfer, and $\bar{l}$ is the mean chord length, which is equal to 2/3 d according to the Cauchy theory [78] if the target is a sphere with diameter d.

Another type is the dose-mean lineal energy, *y_D_*, was calculated as:(3)$$y_{D}=\frac{1}{y_{F}}\int_{0}^{\infty }y^{2}f\left(y\right)\;dy.$$

The incident particle in this study was an electron with energies of 50 eV, 100 eV, 200 eV, 300 eV, 400 eV, 500 eV, 700 eV, 1 keV, 3 keV, 5 keV, 7 keV and 10 keV, with option4 limited to 10 keV. The number of incident particles was set to 10^5^ to minimize the simulation time for energy ≥ 5 keV and 10^6^ for other energy. The targets simulated were spheres with diameters of 2.3 nm, 10 nm and 30 nm, representing the diameters of the B-DNA helix, nucleosome, and chromatin fiber, respectively, according to homogeneity. Moreover, the simulation parameters of previous study [54] were applied, such as without considering the molecule attachment and vibrational excitation processes and considering atomic de-excitation. The target volume is randomly placed at a distance less than the target radius from one energy deposition. 

### 4.4. DNA Damage Model

We examined the physical damage parameters only, not chemical damage, to see how this model performed when the physical damage parameters were changed. In the physical stage of water radiolysis, radiation interacted with DNA molecules and a medium (water). Direct DNA damage was induced from physical interactions between primary and secondary particles within the DNA target. To connect energy deposition close to DNA molecules with DNA damage, mechanistic DNA simulations relied on a DNA damage model. In this work, a single energy threshold (ET) model with 8.22 eV, 12.6 eV, 17.5 eV or 21.25 eV and a linear proportional model between 5 and 37.5 eV were used to predict the damage. The minimum excitation energy of water molecules was 8.22 eV; 12.6 eV was the first ionization energy of water molecules; and 17.5 eV was the energy threshold in KURBUC [39] according to the phenomenological estimation of Charlton [36], as shown in Formula (4):(4)$$P=\left\{\;\begin{array}{c} 1,Edep\geq 37.5\;\mathrm{eV}\\ \frac{Edep\;-\;5}{37.5\;-\;5},5\;\mathrm{eV}<Edep<37.5\;\mathrm{eV}\\ 0,Edep\leq 5\;\mathrm{eV} \end{array}\right.$$

The proportional model, which was also used in PARTRAC [19] and TOPAS-nBio [59,60], assigned a probability of strand breaks of 0 for total energy deposition below 5 eV (based on DNA induction experiments using low-energy photons and electrons [78]), of 1 above 37.5 eV, within a linear interpolation between. Thus, 21.25 eV was the energy deposition value at the probability of 0.5 in Formula (4).

We also investigated the impact of different physics models on physically induced DNA damage, using three recommended physics constructors listed in Table 2 (option2, option4, and option6) and the new physics constructors including the ELSEPA model and option2and4 (Table 1). Additionally, we used 3 bp, 10 bp, 30 bp and 40 bp to test the sensitivity of the DNA damage clustering distance in the simulation. We choose these distances because 10 bp was used most frequently, which is approximately one turn of the double helix, and 3 bp was used by Charlton et al. [35]. A value of 30 bp causes DSBs in a dilute solution at room temperature [79], whereas 40 bp could alter the DNA repair process [58].

The DNA damage classification for the complexity of SSBs and DSBs is shown in Figure 9, based on the classification scheme of Nikjoo et al. [35]. To achieve acceptable statistical accuracy, different runs with different random numbers were performed for some simulated particle energy. The statistical uncertainties of SSB/DSB ratio and the yield of SSBs and DSBs were smaller than 3%.

## 5. Conclusions

DNA damage by 0.5–100 MeV monoenergetic protons irradiation was studied by simulating the yield of DSB and SSB lesions in a chromatin fiber model with the Geant4-DNA toolkit. The default physics constructors in Geant4-DNA including the ELSEPA model and the “option2and4” physics list have been built in our work and were verified with respect to reference data. Our results for the yields of DSBs and SSBs and the ratio of DSBs to SSBs were consistent with other simulations. On the basis of calculated sensitivities, we further estimated the uncertainties of calculated DSB yields and ratio of DSB to SSB due to the uncertainties in the physics constructors, the ET damage model and the clustering distance in the physical stage.

The differences in physics constructors have less influence on the SSB yield and more influence on the DSB yields. Our simulations show that G4EmDNAPhysics_option6 results in the highest DSB yields, whereas the G4EmDNAPhysics_option2 has the lowest DSB yields. The elastic scattering model has little effect on SSB yields, whereas the recommended physics constructors in Geant4-DNA that cause less diffusive distribution result in larger DSB yields due to more concentrated deposited energy.

Differences in the ET damage model can largely affect the DSB yields and the ratio of DSBs to SSBs. For the constant ET model, the DSB yields and the ratio of DSBs to SSBs can decrease up to many times with the ET value from 8.22 to 17.5 eV, whereas the little change for that from 17.5 to 21.25 eV was found in our simulations. The DNA damage for the linear proportional model was consistent with the constant ET model with values between 12.6 and 17.5 eV. Differences in the clustering distance can cause differences up to five times in predicted DSB yields, especially for the clustering algorithm.

However, the pre-chemical and chemical stage and the uncertainly due to the parameters in these stages, such as the reaction radius to search for damage and the temporal length of the chemical stage, were not considered in this work. In addition, the DNA geometry for the human fibroblast was not constructed, and the direct and indirect damage were not distinguished in our simulations. Unfortunately, because of the limitation of the energy for the proton in the Geant4-DNA toolkit (only up to 100 MeV), the energy of proton therapy range from 100 to 230 MeV cannot be simulated. The effect of histone scavenging and base damage will also be investigated in the future. 

## Figures and Tables

**Figure 1 ijms-23-06343-f001:**
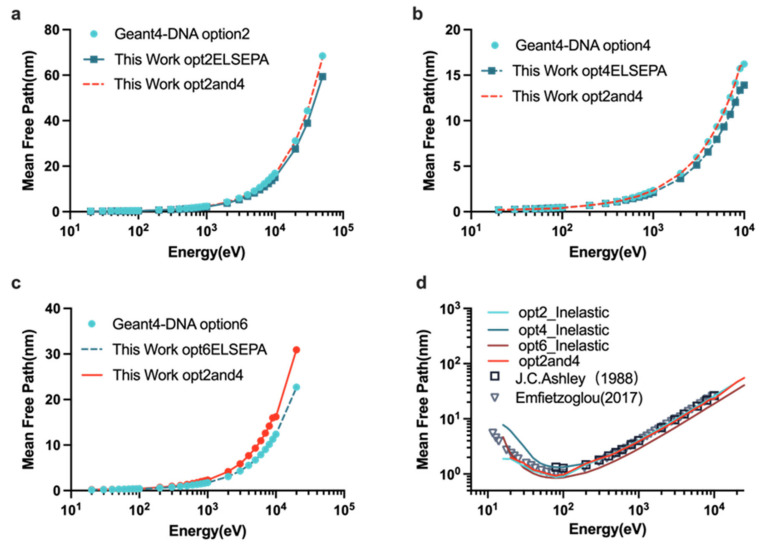
MFP simulations for electrons in liquid water as a function of initial electron energy from 12 eV to 50 keV. (**a**) The total MFP simulations for option2, option2 with ELSEPA and opt2and4. (**b**) The total MFP simulations for option4, option4 with ELSEPA and opt2and4. (**c**) The total MFP simulation for option6, option6 with ELSEPA and opt2and4. (**d**) The inelastic MFP for the option with the ELSEPA model was calculated and compared with the experimental data adapted from Ref. [62] Ashley et al., 1988, and the value adapted from Ref. [63] Emfietzoglou et al., 2017.

**Figure 2 ijms-23-06343-f002:**
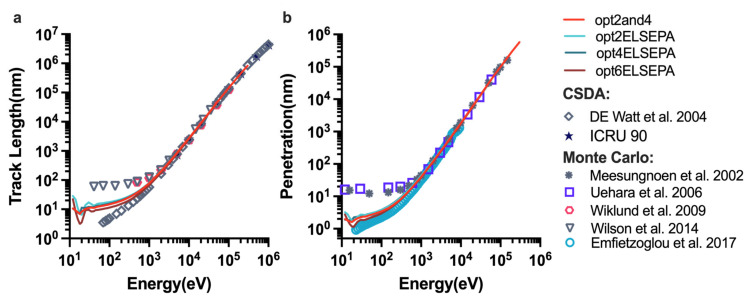
Range simulations in water with different physics constructors as a function of primary electron energy from 10 eV to 1 MeV. (**a**) MCTS simulations of the total path length as a function of incident electron energy compared with the other MC simulations data adapted from Ref. [64] Wiklund et al., 2009, and Ref. [65] Wilson et al., 2014, and continuous-slowing-down-approximation (CSDA) methods data adapted from Ref. [66] Watt et al., 2004, and Ref. [67] Seltzer et al., 2016. (**b**) Variation of the electron penetration range in comparison with data adapted from Ref. [68] Meesungnoen et al., 2002, Ref. [69] Uehara et al., 2006, and Ref. [63] Emfietzoglou et al., 2017.

**Figure 3 ijms-23-06343-f003:**
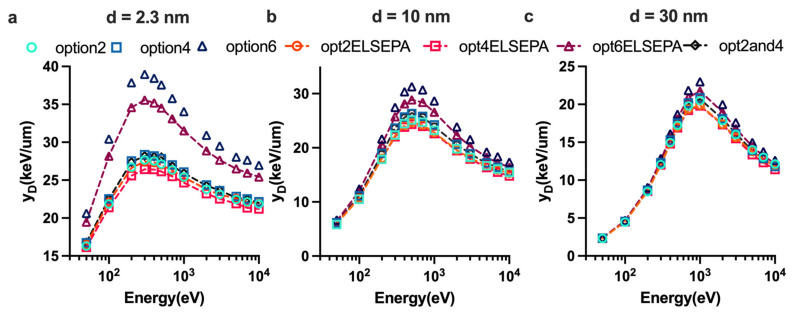
The dose-mean linear energy simulations for different scales of a chromatin simplified model (**a**−**c**).

**Figure 4 ijms-23-06343-f004:**
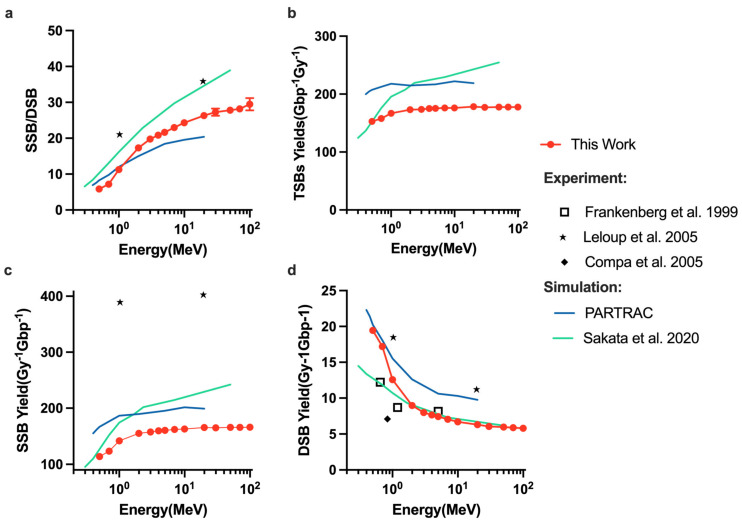
Comparison of results for DNA damages obtained with Geant4-DNA with the experiment data adapted from Ref. [7] Leloup et al., 2005, Ref. [71] Frankenberg et al., 1999, and Ref. [72] Compa et al., 2005, and the simulated data adapted from Ref. [19] Friedland et al., 2010, and Ref. [70] Sakata et al., 2020. All simulations in this work used option2ELSEPA and the linear proportional model. (**a**) The ratio of SSB yield to DSB yield. (**b**) Total strand break (TSBs) yield for protons. (**c**) Total SSB yields for protons. (**d**) Total DSB yields for protons.

**Figure 5 ijms-23-06343-f005:**
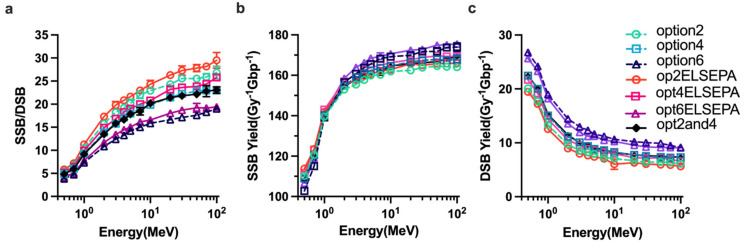
DNA damages simulation with various physics constructors. (**a**) The ratio of SSB yields to DSB yields. (**b**) The yield of SSB for various physics constructors. (**c**) The yield of DSB for various physics constructors.

**Figure 6 ijms-23-06343-f006:**
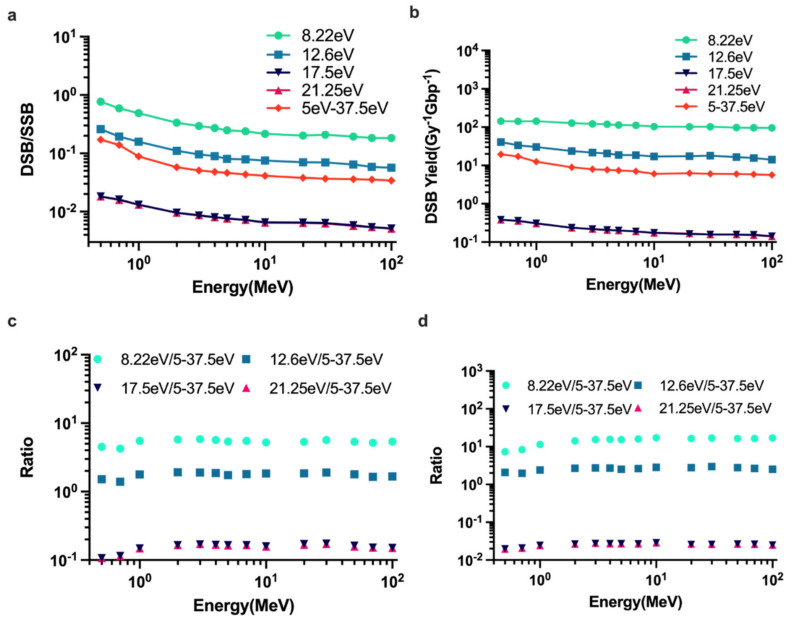
Comparison of DNA damage obtained with different energy threshold model for SSBs. All results were simulated with the opt2ELSEPA model. (**a**) The ratio of DSBs to SSBs with different ET models. (**b**) The total yields for DSBs with different ET models. (**c**) Ratio of constant ET model with linear proportional model of 5−37.5 eV for the DSB/SSB ratio. (**d**) Ratio of constant ET model with linear proportional model of 5−37.5 eV for DSB yields.

**Figure 7 ijms-23-06343-f007:**
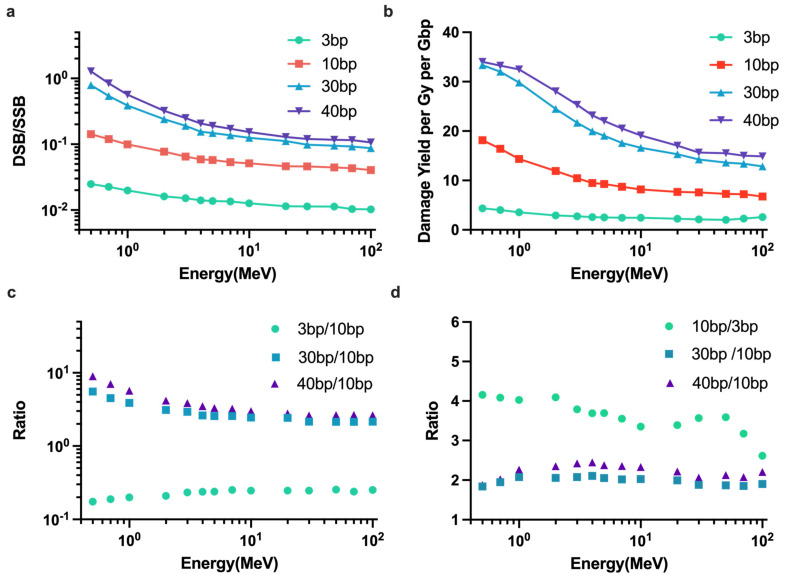
DNA damage simulations with varying damage clustering distance. (**a**) The ratio of the yield of DSBs to the yield of SSBs. (**b**) The yield of DSBs as a function of incident proton energy with damage clustering distance of 3 bp, 10 bp, 30 bp and 40 bp. (**c**) The ratio for DSB/SSB of 10 bp results to other clustering distance used in this work. (**d**) The ratio for DSBs yield of 10 bp to other clustering distance.

**Figure 8 ijms-23-06343-f008:**
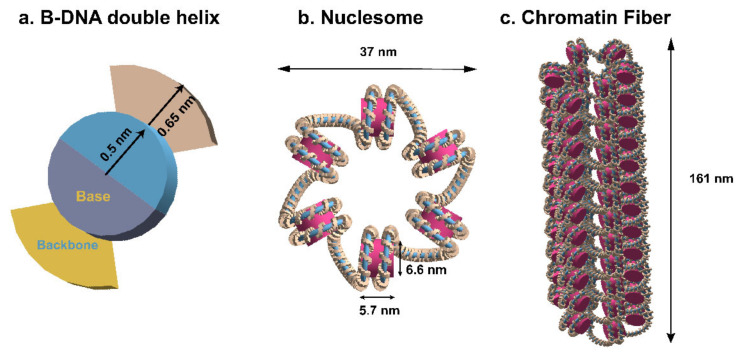
Solenoid chromatin fiber model constructed with Geant4 toolkit. (**a**) Structure of B-DNA double helix. (**b**) Model of six nucleosomes. (**c**) Chromatin fiber geometry.

**Figure 9 ijms-23-06343-f009:**
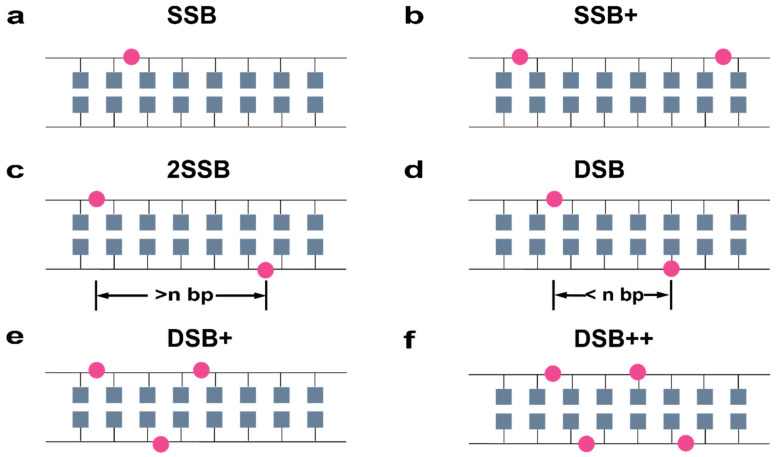
Schematic illustration of damage classification for the complexity of SSBs and DSBs, according to Nikjoo et al., The cube represents bases, the line represents sugar-phosphate backbones, and the circle represents damage points. (**a**) If only one SB occurs in one strand, it is classified as an SSB. (**b**) If more than one SSB occurs in the same strand, they are classified as SSB+. (**c**) If SBs are on both strands and separated by more than n base pair (bp), they are classified as 2 SSB. (**d**) If SBs are on opposite strands and separated by less than n bp, they are classified as a DSB. (**e**) If SBs have the same condition as a DSB and include an additional SSB, they are classified as DSB+. (**f**) If two or more DBSs are present in the segment, the SBs are classified as DSB++.

**Table 1 ijms-23-06343-t001:** Elastic and inelastic models of option8 in Geant4-DNA, the option with the ELSEPA model and the G4EmDNAPhysics_hybrid2and4 and energy limits of applicability.

Process	Physics Constructor				
	Option8	Option2and4	Option2 with ELSEPA	Option4 with ElSEPA	Option6 with ElSEPA
Elastic	CPA100 Model (11 eV–256 keV)	Uehara Model (10 eV–10 keV)	ELSEPA Model (10 eV–1 GeV)	ELSEPA Model (10 eV–1 GeV)	ELSEPA Model (10 eV–1 GeV)
	Champion Model (256 keV–1 MeV)	Champion Model (10 keV–1 MeV)			
Ionization	Born Model (11 eV–1 MeV)	Emfietzoglou Model (10 eV–10 keV)	Born Model (11 eV–1 MeV)	Emfietzoglou Model (10 eV–10 keV)	CPA100 Model (11 eV–256 keV)
		Born Model (10 keV–1 MeV)			
Excitation	Born Model (9 eV–1 MeV)	Emfietzoglou model (8 eV–10 keV)	Born Model (9 eV–1 MeV)	Emfietzoglou Model (8 eV–10 keV)	CPA100 Model (11 eV–256 keV)
		Born Model (10 keV–1 MeV)			

**Table 2 ijms-23-06343-t002:** Physics models of physics lists in Geant4-DNA for electron transport in a liquid water medium and energy limits of applicability.

Process	Physics Constructor		
	Option2	Option4	Option6
Elastic	Champion model	Uehara model	CPA100 model (64)
	(7.4 eV–1 MeV)	(9 eV–10 keV)	(11 eV–256 keV)
Ionization	Born Model	Emfietzoglou model (54)	CPA100 model
	(11 eV–1 MeV)	(10 eV–10 keV)	(11 eV–256 keV)
Excitation	Born Model	Emfietzoglou model	CPA100 model
	(9 eV–1 MeV)	(8 eV–10 keV)	(11 eV–256 keV)
Attachment	Melton model	-	-
	(4–13 eV)		
VibExcitationa	Sanche model	-	-
	(2–100 eV)		

## Data Availability

Not applicable.

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
