# Peer review of "Nanoscale Calculation of Proton-Induced DNA Damage Using a Chromatin Geometry Model with Geant4-DNA"

_ijms, 2022, doi:10.3390/ijms23116343_

Round 1

Reviewer 1 Report

Authors provide interesting study, that would be useful for simulations of precise proton therapy, but I have some concerns about the publication.

Minor:

Authors shoud unite their citations eg, not having number 35 or line 28. It should go either numeric by the occurence in text or alphabetical, quoting Name, Year.

The image quality is terrible. Looks like a low resolution print-screen.

Major:

Line 32 - are there other methods, that would enable that? I could think of few, even though they might not be used in selected experimental publications.

Line 194 - authors should clearly explain what do they mean by "chaning the clustering distance". Is it a parameter in the algorithm? If so, what would be closest to "reality"? Also - how would authors explain that? One does not simply choose the distance of the break in therapy. Is that a flaw of the model? Is it something that is seen experimentally? Even though its partly described in disscussion on lines 250-258, it should be described to greater detail.

In the abstract authors promise comparison with experimental data - they should highlight these comparisons in text, and explain the differences between measured and computed values. Also they should highlight the energies actually used in therapy and see if their computational model and meassured data differ in these regions and suggest for which energy regions are their computations usable.

One of the experimental results come from publication Sakata et. al. 2020, but I have not found it in the literature. I wantetd to see the results from there (eg. if they really only used 2/3 energies in their experiment). If they have used other energies, would it be possible to show them? Or maybe extrapolate to see if the curve would have simmilar shape? Authors should include the publication!

How many times were the simulations done? If once, I would suggest to repeat if it produce same results (at least for some energy points, if the computational time would be excesive). If multiple times, did the simulation bring the exact same datapoints?

Author Response

Point 1: Authors shoud unite their citations eg, not having number 35 or line 28. It should go either numeric by the occurence in text or alphabetical, quoting Name, Year.

Response 1: The full text sorts the references in the order in which they appear.

Point 2: The image quality is terrible. Looks like a low resolution print-screen.

Response 2: All the images containing the data were reproduced using the GraphPad Prism 9 software and the resolution was set to 300dpi. Other images were used Adobe Illustrator and set to 300dpi.

Point 3: Line 32 - are there other methods, that would enable that? I could think of few, even though they might not be used in selected experimental publications.

Response 3: From the publication by F.G. Gabriela(doi:10.3892/ol.2017.6002), strategies for the evaluation of DNA damage in experiment as followings: PCR(Polymerase chain reaction), agarose gel electrophoresis, Ku protein, γH2AX protein, Fluorescence strategies(Comet assay, Alkaline single-cell gel electrophoresis...), Chemiluminescence strategies(Immunohistochemical assay...) and Analytical strategies(Gas chromatography-mass spectrometry).And in this work, we used the Monte Carlo methods to calculate the DNA damage.

Point 4: Line 194 - authors should clearly explain what do they mean by "changing the clustering distance". Is it a parameter in the algorithm? If so, what would be closest to "reality"? Also - how would authors explain that? One does not simply choose the distance of the break in therapy. Is that a flaw of the model? Is it something that is seen experimentally? Even though its partly described in discussion on lines 250-258, it should be described to greater detail.

Response 4: The clustering distance is a parameter in the algorithm as described in the Materials and Methods 4.4( DNA damage model Line ). The clustering distance is the maximum separation between two damage sites on alternative sides of a DNA strand to consider that a DSB has occurred. And the typically value is 10bp which represent the one turn of the double helix. And the discussion on the effect of the clustering distance on lines 276-285.

Point 5: In the abstract authors promise comparison with experimental data - they should highlight these comparisons in text, and explain the differences between measured and computed values. Also they should highlight the energies actually used in therapy and see if their computational model and meassured data differ in these regions and suggest for which energy regions are their computations usable.

Response 5: The comparisons between this simulation and experiment measured by Leloup et al.(https://link.springer.com/article/10.1007/s00411-015-0605-6) were added to the figure 4.The energy used in proton therapy (70MeV-100MeV) were also simulated. Because the limitation of the energy for the proton with Geant4 is 100 MeV, the energy range from 100 MeV to 230 MeV cannot be simulated. And this limit was added in the conclusion on Line 240-266 .

Point 6:One of the experimental results come from publication Sakata et. al. 2020, but I have not found it in the literature. I wantetd to see the results from there (eg. if they really only used 2/3 energies in their experiment). If they have used other energies, would it be possible to show them? Or maybe extrapolate to see if the curve would have simmilar shape? Authors should include the publication!

Response 6: The results from Sakata et al. were simulated not in experiment and we added the experiment and other simulations in figure 4.The references were also included. 

Point 7:How many times were the simulations done? If once, I would suggest to repeat if it produce same results (at least for some energy points, if the computational time would be excesive). If multiple times, did the simulation bring the exact same datapoints?

Response 7: We changed the random seed and simulated several times for some energy points.

Some values were shown in the Table 1-2. The mean of multiple simulations used in this work, and the error was relatively small.

Table 1. The DSB yield of different clustering distance for some energy points

Energy(MeV)

40bp

Mean

SD

0.5

34.0533389

33.9835403

33.999389

34.0121061

0.029876747

4

23.147204

23.311748

23.0666843

23.1752121

0.216432322

Energy(MeV)

30bp

Mean

SD

4

19.9297369

19.81625835

20.32617902

20.02405809

0.21859726

Energy(MeV)

10bp

Mean

SD

20

7.82315051

7.99388294

8.07349283

7.96350876

0.10443423

Table 2. The ratio of DSB yield to SSB yield with different ET model for some energy points

Energy(MeV)

17.5eV

Mean

SD

20

0.00494382

0.00648508

0.00723654

0.00622188

0.000954331

Energy(MeV)

21.25eV

Mean

SD

20

0.00637

0.00649570

0.0057754

0.0062137

0.000384747

Reviewer 2 Report

In this submitted manuscript, Kun Zhu et. al present a study simulating the DNA target response after proton irradiation using Geant4-DNA simulation toolkit. Authors conclude that DNA damage is sensitive to physics constructors, the energy threshold for single strand breaks and the clustering distance for double strand breaks. I do not see this work producing anything more than a nice analogy, perhaps they can improve the abstract, introduction and conclusions to make their goals clearer. In addition, the content of the paper is too redundant, and the key point is not prominent. Here are some other comments:

  • Abstract: “Monte Carlo simulations can quantity various types of DNA damage ….. ”
  • References are not numbered correctly. I see a reference number 1116 in line#37.
  • Line#32: “ …. we used a mechanistic approach with Monte Carlo Track Structure (MCTS) simulation is to quantify the differences in expected DNA damage …” statement is not clear.
  • Introduction seems more like a method section. The introduction should provide relevant background information that describe the depth and challenges of the study.
  • The resolution of the figures is so poor that presented data cannot be distinguished and labels cannot be read.

I my opinion the manuscript does not meet the publication criteria due to a lack of substantial new insight and weaknesses in the presentation.

Author Response

Point 1: In this submitted manuscript, Kun Zhu et. al present a study simulating the DNA target response after proton irradiation using Geant4-DNA simulation toolkit. Authors conclude that DNA damage is sensitive to physics constructors, the energy threshold for single strand breaks and the clustering distance for double strand breaks. I do not see this work producing anything more than a nice analogy, perhaps they can improve the abstract, introduction and conclusions to make their goals clearer. In addition, the content of the paper is too redundant, and the key point is not prominent.

Response 1:The abstract, introduction and conclusion were revised in the new manuscript. The purpose of this work is to accurately calculate the DNA damage with Monte Carlo method. The uncertainty of this method is the DNA scoring model parameters such as the physics constructor, the DNA geometry, the energy threshold for SB. It is an  

Point 2: Abstract: “Monte Carlo simulations can quantity various types of DNA damage ….. ”

References are not numbered correctly. I see a reference number 1116 in line#37.

Response 2: This is an error due to inserting cross-references, which we have corrected in the new manuscript. And all the reference were examined.  

Point 3: Line#32: “ …. we used a mechanistic approach with Monte Carlo Track Structure (MCTS) simulation is to quantify the differences in expected DNA damage …” statement is not clear.

Response 3: This statement was intended to quantify the DNA damage including the calculation of DSB and SSB yields. We have revised this sentence on Line 35

Point 4: Introduction seems more like a method section. The introduction should provide relevant background information that describe the depth and challenges of the study

Response 4: The introduction was revised in the new manuscript. In this work, we used a Monte Carlo toolkit to simulate the early DNA damage. The challenges of the simulation were the precise particle spatial distribution and the uncertainty of the DNA damage calculation model parameters. For the precise particle spatial distribution, the realistic DNA geometry and the physics constructors to simulate the DNA target with the proton were important aspects to the simulation We simulated the DNA damage yield and ratio of DSB to SSB yield with different parameters to understand their influence.

Point 5: The resolution of the figures is so poor that presented data cannot be distinguished and labels cannot be read.

Response 5: All the images containing the data were reproduced using the GraphPad Prism 9 software and the resolution was set to 300dpi. Other images were used Adobe Illustrator and set to 300dpi.

Round 2

Reviewer 1 Report

Authors have greatly improved their paper, which is now in my opinion ready for publication.

I would recommand including the answers to my questions also to the manuscript e.g. the methos of break detection to the introduction and state, that some of the calculations were done repeatedly with negligeble error (maybe have the presented table in suppl.).
